# Total mercury concentrations in invasive lionfish (*Pterois volitans/miles*) from the Atlantic coast of Florida

Eric G. Johnson 👤*◉, Angelina Dichiera👤¤a◉, Danielle Goldberg¤b, MaryKate Swenarton¤c, James Gelsleichter

Department of Biology, University of North Florida, Jacksonville, FL, United States of America

◉ These authors contributed equally to this work.
¤a Current address: Marine Science Institute, The University of Texas at Austin, Port Aransas, TX, United States of America
¤b Current address: Department of Biology and Marine Biology, University of North Carolina at Wilmington, Wilmington, NC, United States of America
¤c Current address: U.S. Fish and Wildlife Service, Anchorage Fish and Wildlife Conservation Office, Anchorage, AK, United States of America
* eric.johnson@unf.edu

**Data Availability Statement:** All relevant data files are available from the Dryad database under the Total mercury concentrations of Florida lionfish

## Abstract

Invasive lionfish (*Pterois volitans/miles*) pose a serious threat to marine ecosystems throughout the western Atlantic Ocean and Caribbean Sea. The development of a fishery for lionfish has been proposed as a strategy for controlling populations; however, there is concern about consumption of this species by humans due to its high trophic position and potential for bioaccumulation of mercury. We analyzed total mercury (THg) in tissues of lionfish from two locations on the east coast of Florida. THg in lionfish increased with size and differed by location and sex. THg was highest in muscle tissue and was strongly positively correlated among tissues. THg in lionfish was lower than other commonly consumed marine fishes, and falls into Florida's least restrictive advisory level. Consumption of lionfish poses a low risk and concerns over mercury bioaccumulation should not present a significant barrier to lionfish harvest.

## Introduction

Biological invasions are a significant and growing threat to global ecosystems causing both ecological harm and staggering economic costs [1, 2]. A marine invader of particular concern in the western Atlantic Ocean is the Indo-Pacific lionfish (a species continuum of two morphologically indistinct species, *Pterois miles* and *P. volitans* or potentially a hybrid of the two; [3]). Following introduction into the coastal waters of south Florida beginning over three decades ago [4–6], a combination of favorable life history traits has enabled this species to become among the most conspicuous and abundant residents in coastal ecosystems throughout the region [7–9].

Lionfish have long venomous spines that deter predation by native predators [10] and are resistant to common parasites [11]; however a recently discovered outbreak of an ulcerative

project at https://doi.org/10.5061/dryad.
p8cz8w9r8.

**Funding:** This work was supported by grants from
the National Science Foundation Research
Experiences for Undergraduates Program (OCE-
1156659; REU - For Students | NSF - National
Science Foundation) to JG, the West Marine -
Marine Conservation Grant Program (BlueFuture -
Grants | West Marine) and University of North
Florida Dean's Leadership Council Faculty
Fellowship Program (UNF - College of Arts &
Sciences - Dean's Leadership Council Faculty
Fellowship Awards) to EGJ, the UNF Department of
Biology (UNF - COAS: Biology - Biology Home) and
UNF Coastal Biology Flagship Program (UNF -
COAS: Biology - about) to AD, and the Guy Harvey
Ocean Foundation to MKS. The results and
interpretations of this work are those of the authors
and do not represent the official views of the NSF,
West Marine, UNF or GHOF. The funders had and
will not have a role in study design, data collection
and analysis, decision to publish, or preparation of
the manuscript.

**Competing interests:** The authors have declared
that no competing interests exist.

skin disease in lionfish in the Gulf of Mexico suggests some susceptibility to disease [12]. Lion-fish also inhabit a wide range of habitats and environmental conditions [13–15], and are capable of long distance dispersal during pelagic egg and larval phases [16, 17] facilitating range expansion and colonization. Lionfish consume a generalist diet and are voracious predators of an array of reef fishes [18, 19]. Lionfish have been shown to reduce native fish recruitment [20] and overall native species biomass [21] in some studies, but not in others [22]. Lionfish directly impact recreationally and commercially important fishes by preying on them as juveniles [19, 23, 24] and impact adults of these same species indirectly through competition for food resources [24, 25].

To mitigate the impacts of lionfish on native ecosystems numerous removal strategies have been proposed [26]. Among these, harvest of lionfish by recreational and commercial spear-fishers has shown promise as one method to help control populations of this invasive species [18, 27–35]. Such efforts are being actively promoted by management agencies throughout the western Atlantic Ocean and Caribbean Sea, but require careful implementation to be most effective [33]. However, although lionfish have a white flesh with flavor and texture similar to highly-valued species such as snapper and grouper; efforts to develop a viable fishery for lion-fish have been hampered by concerns that the concentration of biologically derived toxins, pollutants and heavy metals in this predator may be high enough to present an exposure risk for humans [36]. In Florida, mercury contamination is of particular concern, with high levels of this toxin found in many species of fishes [37–39].

Mercury is a naturally occurring toxic metal that is known to bioaccumulate in the tissue of fishes. As mercury is primarily obtained from the diet, mercury can be magnified in the tissues of aquatic and marine predators, like lionfish, that feed at higher trophic levels [9, 25, 40]. The accumulation of mercury can not only directly impact fish health [41] and adversely affect many aspects of reproduction [42]. Mercury also poses a serious exposure risk for humans who consume fish, particularly for young or pregnant individuals [43–45]. Further, most of the mercury present in fish muscle is present as organic methylmercury (MeHg) [46–48], the most highly toxic and bioactive form [40]. MeHg exposure in humans is almost exclusively from the consumption of fish [49, 50]. To limit human exposure to mercury, dietary guidelines for fish consumption have been established (U.S. EPA, Florida DOH) and because rates of mercury bioaccumulation vary as a function of biological, environmental, and temporal factors, recommended levels of consumption often differ across species, sizes and locations.

The goal of this study was to quantify total mercury (THg) in lionfish as a function of capture location, sex, size and tissue type; information that is critical for evaluating the potential risk to consumers of this species in a rapidly developing fishery. Previous studies have provided assessments for mercury risk in lionfish [51–54]; however, none of these studies were conducted in our region of study. Moreover, only one of the aforementioned studies examined mercury in lionfish across the entire range of sizes currently being harvested [51]. This study builds on earlier work by (1) expanding the spatial coverage to include an assessment of mercury in lionfish from unstudied regions, (2) expanding the range of sizes examined (particularly large individuals which have the highest potential for mercury bioaccumulation), and (3) quantifying mercury levels in different lionfish tissues to assess potential impacts on lionfish health and reproduction.

## Materials and methods

### Ethics statement

All lionfish used in this study were handled in strict accordance with a UNF IACUC protocol (IACUC#13–004) and tissues of opportunity waivers approved by the University of North

Florida. UNF IACUC defines tissues of opportunity as samples collected: (1) during the course of another project with an approved IACUC protocol from another institution; (2) during normal veterinary care by appropriately permitted facilities; or (3) from free-ranging animals by appropriately permitted facilities. Lionfish removals are encouraged by the State of Florida and sample collection locations did not require any specific permissions. No endangered or protected species were harmed during the course of this study.

## Field collections

Lionfish were collected in coordination with four recreational fishing tournaments (also termed derbies): three in northeast Florida (NEF; August 2013, April 2014 and August 2014) and one in southeast Florida (SEF; August 2014; Fig 1). Study regions (Fig 1) are representative, but only approximate, locations of capture since actual coordinates were not provided by tournament fishers. During each derby, teams of recreational divers captured lionfish from local sites and then returned to tournament headquarters where fish were counted, measured and weighed. Lionfish were then separated by location of capture and placed on ice for

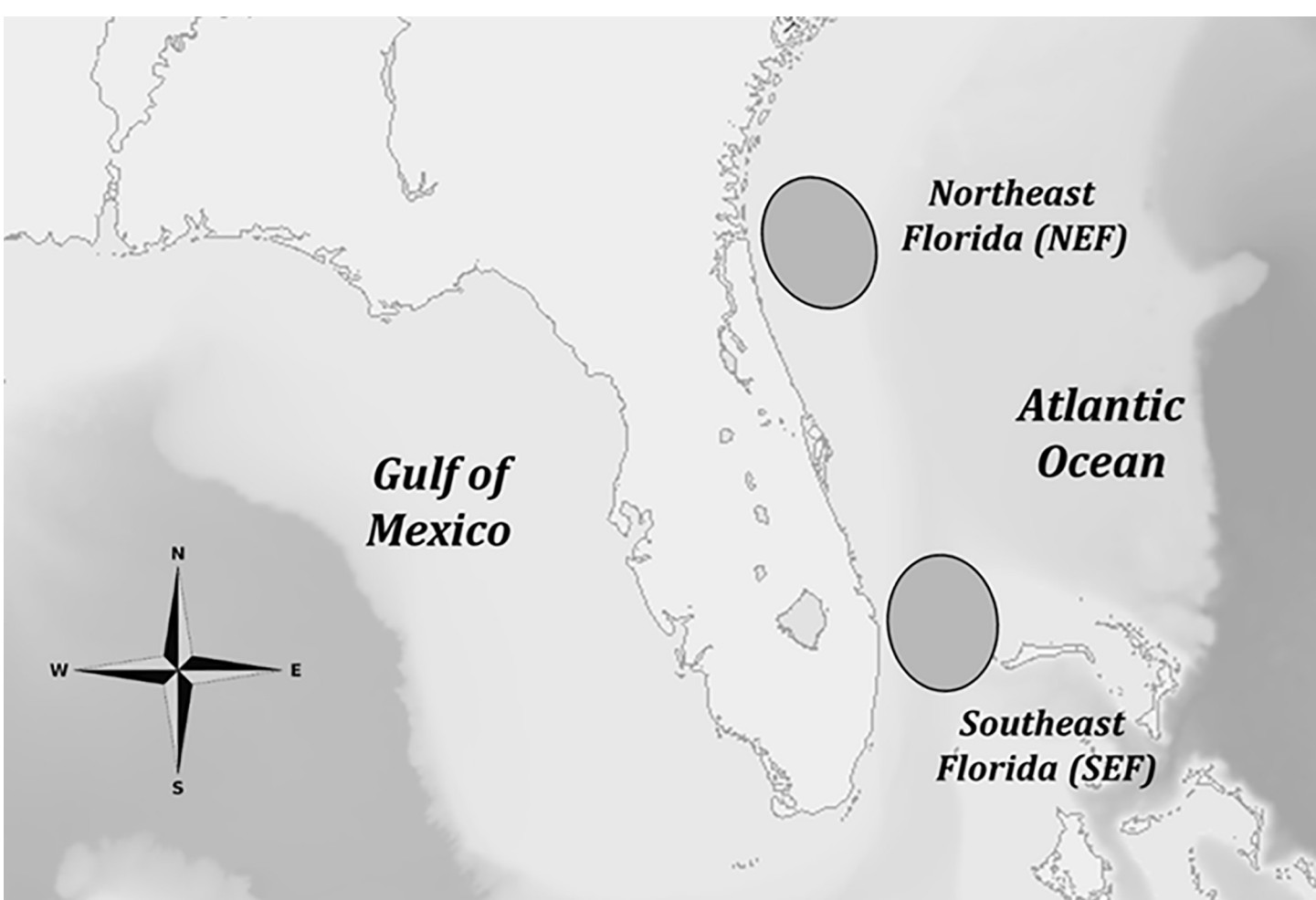

**Fig 1. Map of study locations.** Location of lionfish collection sites from northeast (NEF) and southeast (SEF) Atlantic coasts of Florida. The final figure was modified from a base map created using the USGS National Map Viewer (public domain): http://viewer.nationalmap.gov/viewer/.

**Table 1. Morphometric and THg summary data by location and sex for lionfish in Florida.**

| Location | | Morphometrics | | | THg dw (µg g$^{-1}$) | | | THg ww (µg g$^{-1}$) | | |
|---|---|---|---|---|---|---|---|---|---|---|
| | n | SL (mm) | TL (mm) | Mass (g) | THg (dw) | Min | Max | THg (ww) | Min | Max |
| **A. Northeast Florida** | | | | | | | | | | |
| Male | 50 | 206 ± 45 | 281 ± 57 | 382 ± 231 | 0.27 ± 0.16 | 0.06 | 1.05 | 0.05 | 0.01 | 0.21 |
| Female | 31 | 167 ± 38 | 245 ± 38 | 227 ± 115 | 0.26 ± 0.16 | 0.12 | 0.81 | 0.05 | 0.02 | 0.16 |
| Unknown | 33 | 148 ± 67 | 246 ± 38 | 148 ± 176 | 0.21 ± 0.17 | 0.09 | 0.99 | 0.04 | 0.02 | 0.20 |
| All | 114 | 178 ± 56 | 250 ± 72 | 250 ± 72 | 0.25 ± 0.16 | 0.06 | 1.05 | 0.05 | 0.01 | 0.21 |
| **B. Southeast Florida** | | | | | | | | | | |
| Male | 38 | 178 ± 40 | 267 ± 62 | 285 ± 188 | 0.24 ± 0.26 | 0.11 | 1.30 | 0.05 | 0.02 | 0.26 |
| Female | 22 | 165 ± 27 | 247 ± 38 | 260 ± 148 | 0.32 ± 0.53 | 0.11 | 3.28 | 0.06 | 0.02 | 0.65 |
| Unknown | 1 | 65 | 110 | 12 | 0.17 | 0.17 | 0.17 | 0.03 | 0.03 | 0.03 |
| All | 61 | 167 ± 35 | 167 ± 35 | 167 ± 35 | 0.29 ± 0.44 | 0.11 | 3.28 | 0.06 | 0.02 | 0.65 |
| Total | 175 | 174 ± 50 | 250 ± 66 | 271 ± 202 | 0.26 ± 0.29 | 0.06 | 3.28 | 0.05 | 0.01 | 0.65 |

Mean (± sd) values for sample size (*n*), fish morphometrics: (Standard length (SL), Total length (TL) and Mass (g)), and raw total mercury concentration (µg g$^{-1}$) on a dry weight and wet weight basis for lionfish collected in two regions along Florida's Atlantic coast. THg dry weight was converted to wet weight based on a simple linear regression model (WW = 0.1972*DW); only mean, minimum and maximum values are shown. For each region, data are summarized for all individuals and by sex.

transport to the laboratory. Lionfish were either dissected fresh or frozen whole and stored in freezers at -20˚C until later processing and mercury analysis.

## Sample processing

Lionfish were processed following standard protocols for mercury analysis to minimize the potential for cross-contamination among samples (U.S. EPA, 2000). Each fish was measured for standard length (SL, mm) and total length (TL, mm), weighed (g), and sexed (Table 1). Sex determination was not possible for many smaller immature individuals [29] and not available for a limited number of lionfish (n = 5) which sustained extensive damage during capture and field processing. To represent the portion of the fish most often consumed by humans, muscle was collected from the fillet of the left side of each fish, just above the lateral line [37]. For a subset of male (n = 26) and female fish (n = 31), we collected additional samples of liver, adipose and ovarian tissue for analysis. All samples were dried in a 60˚C oven for 48 hours (a duration sufficient to achieve constant weight of samples) then homogenized using a mortar and pestle prior to analysis of total mercury concentrations (hereafter THg). The proportion of methylmercury (CH$_3$Hg$^+$) was not quantified; however, THg provides a reasonable approximation for finfish muscle in which greater than 95% of total mercury is methylated [47, 55]. Unfortunately, the relationship between methylmercury and THg in other tissues is more variable [56–58] thus THg may not be a good proxy for methylmercury and the concentrations for other tissues in this study are best considered as maximum possible values.

## Mercury analyses

THg in lionfish tissues were measured using a Direct Mercury Analyzer (DMA-80; Milestone Inc., Shelton, CT, USA); a method recognized by the U.S. EPA (EPA Method 7473). The DMA-80 was calibrated weekly using serial dilutions from a liquid standard (1000 mg/L ± 5 mg/L, Certified Reference Material (CRM), ASSURANCE, SPEX CertiPrep, Inc.).

To facilitate comparisons with previous work and to evaluate mercury levels relative to national [44] and local [59] guidelines for consumption, we converted THg in muscle tissue from a dry weight to wet basis using a simple linear regression model fit to a subsample of fish for which sample weights were available both before and after drying. The slope of the line

allowed for percent moisture to be calculated and THg dry weight to be converted to a wet weight basis for unknown samples.

Quality assurance and control (QA/QC) protocols were rigidly followed to ensure acceptable levels of accuracy and precision in the data. Weekly calibration of the DMA-80 with liquid mercury standard yielded standard curves with $r^2$ values > 0.99. Method blanks returned THg concentrations well below the lowest recorded value from any fish sample in the study (<10% of the lowest value recorded in the study; EPA method 7473) confirming that samples were not substantially affected by contamination. Standard blanks using a solid standard (CRM, NIST Standard Ref 2709a) were all within expected values, $0.9 \pm 0.2$ μg g$^{-1}$. The solid standard was chosen because no fish tissue standard was available at the time of analysis; however subsequent paired analyses of the solid standard and fish protein standard (DORM-4, National Research Council of Canada) using the same protocol demonstrated that both standards fell within their respective expected ranges. Duplicate (n = 66) and triplicate (n = 34) tissue samples from the same individual were run a minimum of every 10 samples over the course of the study and yielded coefficients of variation (CVs) averaging 2.17% indicating acceptable levels of precision.

## Statistical analysis

All statistical analyses were run within SAS 9.4 (SAS Institute), SPSS Statistics 25 for Windows (IBM) or SigmaPlot (Systat Software) software packages. When necessary THg data were log-transformed (ln THg) prior to analyses to satisfy the assumptions of normality (Kolmogorov-Smirnov test) and homogeneity of variances (Levene's test) required for parametric statistical tests. Temporal differences in THg in lionfish from NEF were assessed using a one-way analysis of covariance (ANCOVA) with collection date as a main factor and standard length (SL) as a covariate. SL was chosen as the best covariate for all analyses because body mass was not available for all fish and TL is more likely to be biased by potential damage to the caudal fin occurring naturally or during capture, storage and processing. Potential differences in THg between locations (NEF and SEF) and sexes (male and female) were tested using a two-way ANCOVA model with SL as a covariate. In cases where interactions were present between model main effects and covariates, we applied the Johnson-Neyman procedure as suggested by Wilcox [60].

Differences in THg between sexes and among tissues were assessed using a two-way repeated measures ANCOVA (RM ANCOVA) for three tissues (muscle, liver, adipose) excluding ovarian tissue. A one-way RM ANCOVA model was also run for females separately to include ovary tissue in the analysis. In all cases, we accounted for violations of the assumption of sphericity by adjusting model degrees of freedom [61]. Following significant results from ANCOVA, a post-hoc multiple comparison test (Dunn-Šidák, [62]). was applied to determine significant differences among tissues. The relationship among THg in various tissues was assessed using correlation (Pearson's *r*) for females and males independently. To maintain experimentwise error rate (nine total comparisons), we applied the sequential Bonferroni correction when assessing significance [63].

## Results

A total of 175 samples of lionfish muscle tissue were analyzed for THg. The samples included 88 males, 53 females and 34 individuals for which sex could not be determined. Overall, fish ranged widely in length (SL: 57 to 306 mm, x̄ = 175 mm, TL: 83 to 411mm, x̄ = 250 mm), weight (4.5 to over 1000g, x̄ = 271g) and THg (0.01 to 3.28 μg g$^{-1}$ dw; x̄ = 0.26; Table 1).

To compare our results to earlier work and to recommended guidelines for consumption, we converted THg in muscle tissue from a dry weight to wet weight basis using a simple linear regression model. The linear relationship between THg wet weight and dry weight (WW = 0.19*DW) was highly significant ($r^2$ = 0.99, p <0.0001); no intercept was included in the final linear model since it was not significantly different from 0 ($t$ = 0.002, p = 0.998). The percent moisture in lionfish muscle averaged 80.4 ± 1.8% with high precision (CV = 0.02) and was independent of fish size, sex, and location. Converted THg on a wet weight basis is provided in Table 1.

Collection time had no significant effect on THg ($F_{[2,57]}$ = 1.05, p = 0.37), thus data for NEF were pooled across time for subsequent analyses. Our initial two-way ANCOVA model revealed several significant interactions (p <0.05) among location and SL (Table 2A). To aid with interpretation of the data and to remove potentially confounding interactions, we ran ANCOVA models for each location separately (Table 2B and 2C). In these reduced models, ANCOVA revealed no significant differences in THg by sex at either site (Table 2B and 2C, Fig 2), although females had generally higher THg for a given size. A one-way ANCOVA model with sexes pooled revealed a highly significant positive relationship between SL and THg and a significant effect of location (Table 2D, Fig 3). Additionally, a significant interaction indicated that the regression slopes for the two locations were not the same (Table 2B). Further analysis (Johnson-Neyman procedure; [60]) determined that individuals greater than 196 mm

**Table 2. Analyses of covariance (ANCOVA) tables for ln total mercury (µg g$^{-1}$) concentrations in lionfish muscle tissue.**

| A. Full model | | | | | |
|---|---|---|---|---|---|
| Source | df | SS | MS | F | p |
| Location | 1 | 0.94 | 0.94 | 6.97 | 0.009 |
| Sex | 1 | 0.09 | 0.09 | 0.66 | 0.417 |
| SL | 1 | 21.64 | 27.69 | 149.22 | 0.000 |
| Location × Sex | 1 | 0.44 | 0.44 | 5.51 | 0.074 |
| Location × SL | 1 | 1.08 | 1.08 | 10.57 | 0.005 |
| Sex × SL | 1 | 0.97 | 0.97 | 5.83 | 0.058 |
| Location × Sex × SL | 1 | 0.44 | 0.44 | 5.41 | 0.073 |
| Error | 132 | 17.73 | 0.13 | | |
| **B. Northeast Florida** | df | SS | MS | F | p |
| Sex | 1 | 0.09 | 0.09 | 0.89 | 0.349 |
| SL | 1 | 9.94 | 9.94 | 95.94 | 0.000 |
| Sex × SL | 1 | 0.00 | 0.00 | 0.01 | 0.922 |
| Error | 77 | 7.98 | 0.10 | | |
| **C. Southeast Florida** | df | SS | MS | F | p |
| Sex | 1 | 0.36 | 0.36 | 2.01 | 0.162 |
| SL | 1 | 12.05 | 12.05 | 119.68 | 0.000 |
| Sex × SL | 1 | 0.71 | 0.71 | 3.94 | 0.052 |
| Error | 55 | 9.76 | 0.18 | | |
| **D. Sexes combined** | df | SS | MS | F | p |
| Location | 1 | 2.31 | 2.31 | 12.72 | 0.000 |
| SL | 1 | 16.54 | 16.54 | 91.02 | 0.000 |
| Location × SL | 1 | 2.47 | 2.47 | 13.57 | 0.000 |
| Error | 171 | 22.01 | 0.17 | | |

Output from Analyses of covariance (ANCOVA) models of ln total mercury (µg g$^{-1}$) concentrations in lionfish muscle tissue from A. Both locations, B. Northeast Florida (NEF), and C. Southeast Florida (SEF) D. Sexes combined.

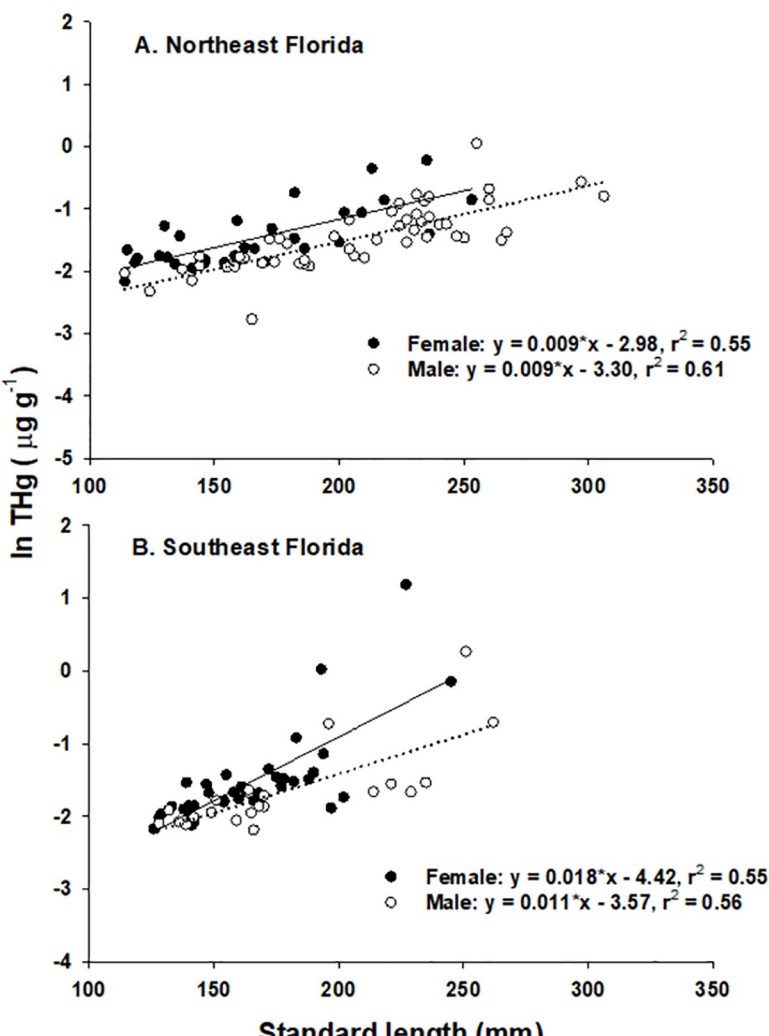

**Fig 2. Relationships between lionfish and THg concentration by sex.** Linear regression relationships between size (Standard length) and total mercury concentration for female (solid circles, solid line) and male (open circles, dotted line) lionfish in northeast Florida (A) and southeast Florida (B).

SL were significantly higher in THg in SEF than in NEF, while smaller individuals were not significantly different between locations (Fig 3).

THg varied significantly among tissues ($F_{1.2, 66.7}$ = 9.36, n = 78, p = 0.002), but did not vary between sexes ($F_{1,53}$ = 1.81, n = 78, p = 0.18; Fig 4), nor was an interaction present. For both sexes, muscle had significantly higher THg than other tissues (muscle > liver > adipose; Dunn-Šidák, Fig 4). In females, THg in ovaries was significantly different from muscle and adipose, but not liver (Dunn-Šidák, Fig 4). THg in all tissues from females were highly correlated (r from 0.66 to 0.88, p < 0.001, Fig 5A–5F); however only THg in muscle and liver tissues were correlated in male lionfish (r = 0.95, p < 0.001, Fig 5I).

## Discussion

The main findings of this study were: (1) mercury concentrations in invasive lionfish from northeast and southeast FL were similar to previously reported values and generally below comparable values for commonly consumed reef species, (2) muscle had significantly higher

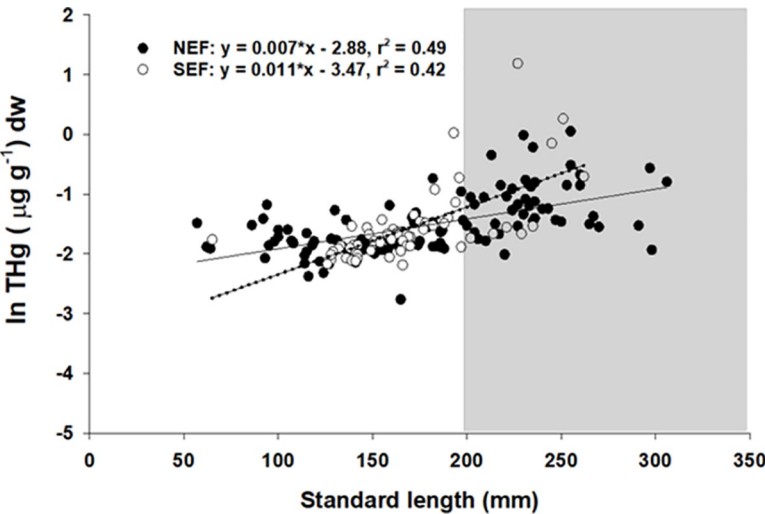

**Fig 3. Relationships between lionfish and THg concentration by study location.** Linear regression relationships between size (Standard length) and total mercury concentration in lionfish from NEF (solid circles, solid line) and SEF (open circles, dotted line). The shaded region indicates the sizes (SL > 196 SL) for which lionfish THg was significantly higher ($\alpha$ = 0.05) in SEF than in NEF.

levels of mercury than other tissues or organs, (3) mercury levels increased with size and rates of accumulation differed by location, (4) females were generally higher in mercury than males for a given size, and (5) mercury concentrations in all lionfish tissues examined in this study were highly correlated for females, but only between liver and muscle for males. Collectively, our findings indicate that THg in lionfish is low and lionfish are safe for human consumption.

## THg in lionfish from northeast and southeast FL

Mean THg in lionfish (0.05–0.06 $\mu g\ g^{-1}$ ww; Table 1) in our study were within the range previously reported from Florida (0.02–0.15 $\mu g\ g^{-1}$ ww; [52, 53]). Our results were also within the

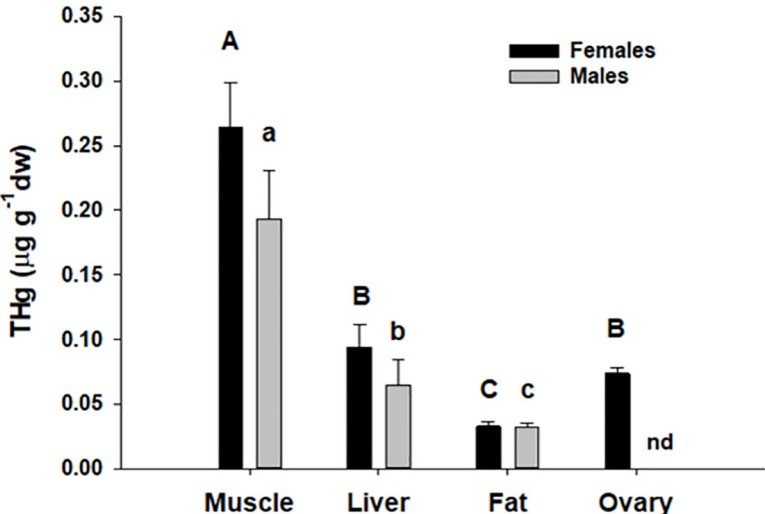

**Fig 4. Comparison of THg concentrations in lionfish tissues by sex.** Total mercury concentrations in lionfish tissues in female (solid black bars) and male (solid grey bars) lionfish. Male testes were not sampled (nd). No significant differences were observed for tissues as a function of sex; letters (capitals = females; lower case = males) above tissues represent significant differences from total mercury in tissue types from a Ryan's Q post-hoc test (see text for details) following ANCOVA.

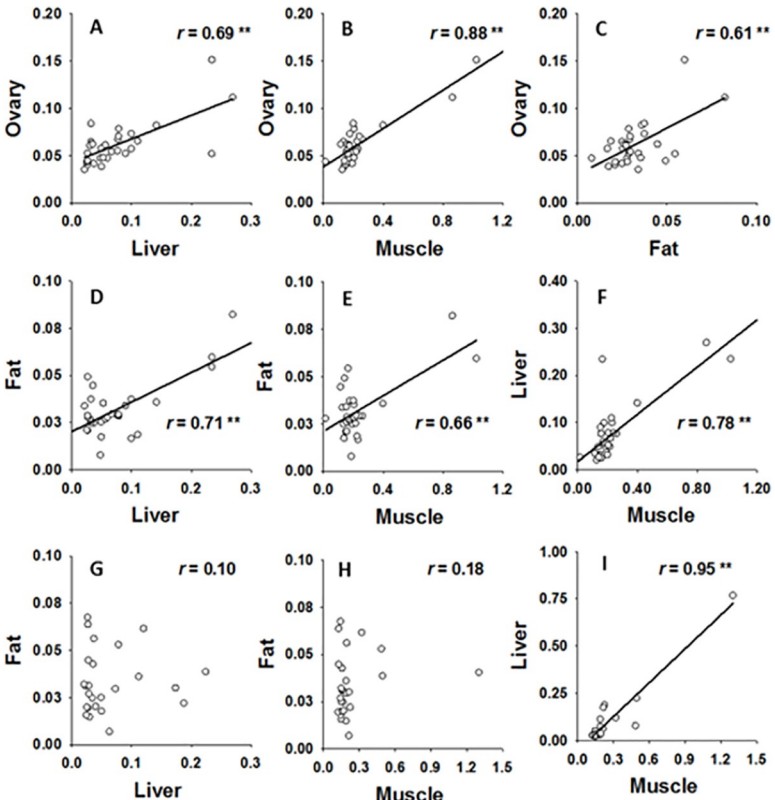

**Fig 5. Relationships in THg concentrations among lionfish tissues.** Relationship among THg (all axes are shown in μg g$^{-1}$ dw) in various tissues in female (Panels A-F, n = 31) and male lionfish (Panels G-I, n = 26). Coefficients from Pearson canonical correlation analysis (*r*) are inset within each Panel, significant correlations at α = 0.05 with sequential Bonferroni correction are noted by asterisks (**).

range of lionfish THg studied in Jamaica (0.016–0.061 μg g$^{-1}$ ww; [51]) and Curaçao (Range: 0.008–0.106 μg g$^{-1}$ ww; [54]).

Overall, THg in lionfish was lower than native predatory reef fishes that occupy similar trophic positions in offshore hard bottom habitats like those sampled in our study. THg in carnivorous fishes of similar size (e.g., red grouper, *Epinephelus morio*; black sea bass, *Centropristis striata*) are 3–4 times higher than the values observed in our study [37, 64] and more than 8–10 times higher in larger, long-lived species such as gag (*Mycteroperca microlepis*) and red snapper (*Lutjanus campechanus*, [64, 65]. The presence of low THg in lionfish is somewhat surprising based on studies of diet [9, 19, 23, 66] which indicate that lionfish are carnivores throughout their life history and almost exclusively piscivorous as adults. Indeed, stable isotope analysis paired with diet analyses suggests that both small (presumably young) and larger lionfish are feeding at the same trophic level [54]. Food web modeling and stable isotope analysis also demonstrates a high trophic position for lionfish within the invaded food web and a large degree of overlap with native top predators [24, 25, 67]. Thus, low levels of THg in lionfish do not appear to be related to differences in feeding ecology as has been demonstrated in groupers and sea basses from this region [65]. Low levels of THg in lionfish could reflect species-specific physiological attributes that favor either reduced uptake or increased rates of elimination. Although lionfish physiology is poorly understood; these rates are highly variable across taxa [68] and in fishes [69].

Low THg in lionfish may be at least partially explained by growth dilution, given the extremely rapid growth rates of lionfish relative to native predators [64]. Lionfish grow to nearly 300 mm TL by age two [27, 70–72]; while important fishery species from Florida waters such as red grouper [73], black sea bass [74], and grey snapper achieve sizes of 200, 127, and 204 mm TL at age two, respectively. As a result, young lionfish (1–3 years old) of similar size to these species have been accumulating THg for as little as half the time. This rationale is supported by empirical studies for many fishes for which age is a better predictor of THg than body size or mass [75, 76].

## Relationship between THg, size and location

Because THg is not readily depurated, it accumulates within tissues over time and the amount of THg in fish tissues is expected to increase as fish age [65, 77]. Our data are in close agreement with this commonly observed pattern; THg in lionfish was significantly positively correlated with fish length (a proxy for age) accounting for between 50 and 60% of the variation ($r^2$ = 0.55–0.61) in THg depending on sex and location of capture (Fig 2A and 2B). Rates of THg bioaccumulation were site-specific and significantly higher in SEF than in NEF resulting in significantly mercury levels for the largest fish (SL > 196mm; Table 2D, Fig 3). Higher concentrations of THg in SEF could be linked to ambient oceanographic conditions such as warmer ocean temperatures, which have been shown to increase methylation rates by marine primary producers [78]. Previous work has noted much larger differences among regions in Florida [53]; however that study examined mercury at a larger spatial scale, in both nearshore and offshore sites, and sampled in locations with both historic point sources (medical waste incineration) and biogeochemical factors that result in enhanced mercury methylation and bioavailability [79–81] distinct from our offshore collection sites. In particular, lionfish from the Florida Keys have been shown to be high in mercury [53], a pattern consistent with elevated mercury in other studies [52] and in other piscivorous fishes from that region [37, 41, 82].

Both the positive relationship between THg and size in lionfish and site-specific bioaccumulation rates have been reported previously [53], but the strengths of the relationships were weaker and more variable ($r^2$ = 0.07–0.35) than reported here. The reduced variability observed in our study may result from our regional spatial scale and similar habitats (offshore hard bottom) in comparison with the earlier study which sampled a larger geographic area, at nearshore and offshore locations, and sampled in the Florida Keys which has unique biogeochemical properties due to the nearby Everglades watershed [53, 83].

## Comparison of THg in lionfish tissues

The differential uptake and accumulation of THg within the body tissues of fishes is not well resolved. While numerous studies have reported low levels of THg in muscle relative to lipophilic tissues such as liver [84], others have found no difference [85, 86] or elevated THg in muscle [85, 87–89]. In this study, THg in lionfish muscle was significantly higher than in other tissues. This finding is in agreement with earlier work [90–92] which supports a general pattern of higher THg in muscle relative to internal organs in fish from lightly contaminated localities which is likely the case with our fish collected in offshore marine waters (>20 km from land). A key limitation of the current study was that methylmercury was not quantified. While numerous studies indicate that THg is a good proxy for methylmercury in fish muscle [46–48, 55], the fraction of the THg pool present as methylmercury is highly variable in other tissue types [58], can vary as a function of fish size [57] and among species [47, 93]. Future work on Hg speciation in lionfish organ tissues is needed to assess the value of using lionfish

as an indicator species to assess ecosystem health and risk (e.g., [91]), and for better understanding ecophysiological mechanisms underlying Hg distribution, detoxification, and sequestration in this species [94]. Despite these limitations, THg represents the maximum possible concentration for methylmercury and observed THg in all tissues were below accepted thresholds for negative health effects [95] and reproductive impairment [96]. Thus, lionfish appear unlikely to be substantially affected by mercury toxicity.

THg in lionfish were generally positively correlated among tissues as is commonly seen in fishes [94]. However, while this relationship was particularly strong for females; only liver and muscle tissue were found to be correlated in males. This finding could indeed reflect real sex-specific differences in physiology leading to differential accumulation and sequestration of THg in fat tissue; however further study will be required to determine if this trend is real and, if so, the physiological processes that underlie it.

## Implications for human health

In the present study, THg in lionfish captured from the east coast of Florida were low (0.05 $\mu g\ g^{-1}$), placing them in Florida's least restrictive consumption advisory level [59] and within the range of fishes (e.g., salmon, tilapia, cod) promoted for safe consumption by the EPA-FDA [44]. Because lionfish are a marine species found in the southeastern U.S. and Caribbean, they are most likely to replace similar regional species such as grouper and snapper in the diet. THg in lionfish is lower than other commonly consumed reef fishes of similar size such as red grouper, (*Epinephelus morio*, 0.17 $\mu g\ g^{-1}$ ww), grey snapper (*Lutjanus griseus*, 0.18–0.21 $\mu g\ g^{-1}$ ww), graysby (*Cephalopholis cruentata*, 0.16 $\mu g\ g^{-1}$ ww), and black sea bass (*Centropristis striatus*, 0.14 $\mu g\ g^{-1}$ ww; [37, 65] and much lower than fishery legal-sized individuals of larger species such as gag (*Mycteroperca microlepis*, 0.40 $\mu g\ g^{-1}$ ww), black grouper (*M. bonaci*, 0.91 $\mu g\ g^{-1}$ ww, and red snapper (*L. campechanus*, 0.49 $\mu g\ g^{-1}$ ww; [64, 65]. Thus, lionfish would appear to represent a low mercury alternative to these species, many of which have been severely depleted by commercial and recreational fishing pressure.

THg in lionfish increased with size, a general pattern consistently observed in fishes. One of the largest fish (SL > 300) in our study had THg concentrations that would fall into the FDOH limited consumption (0.5–1.5 ppm) category which calls for restricting consumption to once a month for women of childbearing age and children and weekly for all others [59]. However, these large fish are exceptionally rare; lionfish harvested in Florida are predominantly young fish (SL < 250 mm); larger fish (SL > 250) comprised only 5% of individuals in northeast Florida [71].

Overall, lionfish yields a comparable amount of flesh to similar-sized marine food fishes, has high levels of fatty acids beneficial for health, and fares favorably in direct comparisons with other high value marine species [97]. Our findings indicate that levels of THg in lionfish are low and lionfish are safe for human consumption. As such, concerns over THg in lionfish should not present a significant roadblock to the continued development of directed commercial and recreational fisheries for this invasive species.

## Acknowledgments

We would like to thank the recreational and commercial spearfishers who collected the lionfish without whom this study would not have been possible. Assistance with sample collection and processing was given by numerous undergraduate and graduate students, in particular Mickhale Green and Corey Corrick. We also thank Chris Guppenberger for initial setup and ongoing technical assistance with the DMA-80 mercury analyzer, and Dr. Amy Lane for creating the liquid mercury standard solutions for calibrations. Samantha Ehnert provided

extensive training to numerous undergraduate students and helped to maintain and trouble-shoot the DMA-80 over the course of the study duration.

## Author Contributions

**Conceptualization:** Eric G. Johnson, Angelina Dichiera, Danielle Goldberg, MaryKate Swenarton.

**Data curation:** Eric G. Johnson, Angelina Dichiera, Danielle Goldberg.

**Funding acquisition:** Eric G. Johnson, MaryKate Swenarton, James Gelsleichter.

**Investigation:** Eric G. Johnson, Angelina Dichiera, Danielle Goldberg, MaryKate Swenarton, James Gelsleichter.

**Methodology:** Eric G. Johnson, Angelina Dichiera, MaryKate Swenarton, James Gelsleichter.

**Project administration:** Eric G. Johnson.

**Resources:** Eric G. Johnson.

**Software:** Eric G. Johnson.

**Supervision:** Eric G. Johnson.

**Validation:** Eric G. Johnson.

**Visualization:** Eric G. Johnson.

**Writing – original draft:** Eric G. Johnson.

**Writing – review & editing:** Eric G. Johnson, Angelina Dichiera, Danielle Goldberg, Mary-Kate Swenarton, James Gelsleichter.

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
