## [Decision Letter · Decision Letter 0]

8 Jul 2020

PONE-D-20-15978

Total mercury concentrations in invasive lionfish (Pterois volitans/miles) from the Atlantic coast of Florida

PLOS ONE

Dear Dr. Johnson,

Thank you for submitting your manuscript to PLOS ONE. After careful consideration, we feel that it has merit but does not fully meet PLOS ONE’s publication criteria as it currently stands. Therefore, we invite you to submit a revised version of the manuscript that addresses the points raised during the review process.

We look forward to receiving your revised manuscript.

Kind regards,

Myra E Finkelstein

Academic Editor

PLOS ONE

Additional Editor Comments:

In your revision please make sure to fully address each of the reviewers' comments.

In particular for reviewer 1, make sure to address this general criticism of the paper:

"But unfortunately measuring only total Hg does not allow for new insights into the disposition of mercury. And no attention is paid to possible sources of Hg that may explain the observed regional differences. In the introduction broad sweeping objectives are presented for this project, but almost none of those are addressed in the results and discussion." and for Reviewer 2 please indicate why a fish tissue SRM was not used.

Journal Requirements:

2. In your Methods section, please provide additional location information of the study sites, including geographic coordinates for the data set if available.

4. We note that Figure 1 in your submission contain map images which may be copyrighted. All PLOS content is published under the Creative Commons Attribution License (CC BY 4.0), which means that the manuscript, images, and Supporting Information files will be freely available online, and any third party is permitted to access, download, copy, distribute, and use these materials in any way, even commercially, with proper attribution. For these reasons, we cannot publish previously copyrighted maps or satellite images created using proprietary data, such as Google software (Google Maps, Street View, and Earth). For more information, see our copyright guidelines: http://journals.plos.org/plosone/s/licenses-and-copyright.

4.1.    You may seek permission from the original copyright holder of Figure 1 to publish the content specifically under the CC BY 4.0 license.

4.2.    If you are unable to obtain permission from the original copyright holder to publish these figures under the CC BY 4.0 license or if the copyright holder’s requirements are incompatible with the CC BY 4.0 license, please either i) remove the figure or ii) supply a replacement figure that complies with the CC BY 4.0 license. Please check copyright information on all replacement figures and update the figure caption with source information. If applicable, please specify in the figure caption text when a figure is similar but not identical to the original image and is therefore for illustrative purposes only.

Reviewers' comments:

Reviewer's Responses to Questions

**Comments to the Author**

1. Is the manuscript technically sound, and do the data support the conclusions?

Reviewer #1: Partly

Reviewer #2: Yes

Reviewer #3: Yes

2. Has the statistical analysis been performed appropriately and rigorously? 

Reviewer #1: Yes

Reviewer #2: Yes

Reviewer #3: Yes

3. Have the authors made all data underlying the findings in their manuscript fully available?

Reviewer #1: Yes

Reviewer #2: No

Reviewer #3: Yes

4. Is the manuscript presented in an intelligible fashion and written in standard English?

Reviewer #1: Yes

Reviewer #2: Yes

Reviewer #3: Yes

5. Review Comments to the Author

Reviewer #1: This manuscript describes the concentrations of total mercury in lionfish from 2 locations off the Atlantic coast of Florida.

The study appears to be well executed, but is very narrow in its conclusions, and does not add any new insights to the scientific knowledge about mercury accumulation in fish species. The results are relevant for the developing market for this species, confirming what others have published already. But unfortunately measuring only total Hg does not allow for new insights into the disposition of mercury. And no attention is paid to possible sources of Hg that may explain the observed regional differences. In the introduction broad sweeping objectives are presented for this project, but almost none of those are addressed in the results and discussion.

Some detailed comments:

Line 52: a more accurate statement is that in the SE USA there is a species continuum for these two species, or even just one hybrid species (see Wilcox CL, Motomura H, Matsunuma M, Bowen BW. Phylogeography of Lionfishes (Pterois) Indicate Taxonomic Over Splitting and Hybrid Origin of the Invasive Pterois volitans. J Hered. 2018 Feb 14;109(2):162-175). Based on my own observations it is difficult to separate the two species on morphological characteristics, and in one location individuals with characteristics of both species can be found.

Line 59: recently a disease outbreak has shown that the species is not all that resistant (Precipitous Declines in Northern Gulf of Mexico Invasive Lionfish Populations Following the Emergence of an Ulcerative Skin Disease. Harris HE, Fogg AQ, Allen MS, Ahrens RNM, Patterson WF 3rd.Sci Rep. 2020 Feb 4;10(1):1934. doi: 10.1038/s41598-020-58886-8)

Line 65/66: However, others have found no significant effects of lionfish on native fish populations (Invasive lionfish had no measurable effect on prey fish community structure across the Belizean Barrier Reef. Hackerott S, Valdivia A, Cox CE, Silbiger NJ, Bruno JF. PeerJ. 2017 May 25;5:e3270. doi: 10.7717/peerj.3270. eCollection 2017.)

Line 72: however, it is doubtful that spear fish derbies will have a significant effect on populations size (Effectiveness of removals of the invasive lionfish: how many dives are needed to deplete a reef? Usseglio P, Selwyn JD, Downey-Wall AM, Hogan JD. PeerJ. 2017 Feb 23;5:e3043. doi: 0.7717/peerj.3043. eCollection 2017)

Line 87: I would say “higher trophic levels” because lionfish are not top predators like sharks, and there food is in general smaller fish.

Line 101 and further: these numbered aims are laudable, but they are hardly at all discussed in the discussion. For instance: there are several papers on lionfish as biomonitoring species for other chemicals, but none of them is discussed. Aims 2 and 3 are also not discussed, only 4.

Line 111: Do you really think that Hg in fish is linked to “high coastal human population density”? For as far as I know, Hg sources along the East Coast are mostly atmospheric deposition of elemental Hg, getting in the atmosphere through coal-fired power plants. In coastal anoxic marsh sediments this elemental Hg is turned into organic Hg, which is bioavailable. In the discussion it is suggested that the Everglades are a source of Hg in fish in the Keys, which makes sense to me. But why would atmospheric deposition in North Florida be different from the South? Where are the coal fired power plants located in FL?

Line 148: I don’t think you need to state that only females have ovaries

Line 236/237: if you set your significance level a 5%, and there are no significant differences between the sexes, then don’t say that the females are generally higher because it is not significant!

Line 284: point 4) see above: if your statistical analysis says there are no differences, then there are no differences!

Line 316-323: this is called growth dilution

Line 337/338: it would be nice to just mention here what these “known processes” are!

Line 363: I fully agree with this statement!

Reviewer #2: *3. Have the authors made all data underlying the findings in their manuscript fully available?

I only said no because on the PDF I received the statement as to where the data can be found is incomplete: "All data files are available from the University of North Florida database (accession number(s) XXX, XXX.)" X's need to be replace with numbers for reader to be able to access. (Maybe they will be on publication, if so, disregard this comment.)

4. Is the manuscript presented in an intelligible fashion and written in standard English?

I answered 'yes' because the edits are minimal. Please see the following comments broken down line by line for typos to fix and minor edits for sentence clarity.

Comments to author:

I realized a bit too late that I wasn’t supposed to provide copyediting along with other review. I am still going to share some details I noticed because I hope they will be useful in assembling the final manuscript. I would like to see this paper published so that others can access these results for comparison and for public knowledge. I appreciate the useful impacts of this study for invasive species control, human health knowledge, and for promoting the idea that lionfish could be consumed instead of other imperiled species.

Line 54 comma after Hamner et al.

Line 63 add comma after Johnston et al.

Line 64 add comma after Cote et al.

Line 65 add comma after Peake et al.

Line 68 add comma after Chagaris et al.

Line 75 add comma after Dahl et al.

Line 88-92 The accumulation of mercury can directly impact fish health (Adams et al., 2010) and can adversely affect many aspects of reproduction (Weis, 2009). Mercury also poses a serious exposure risk for humans who consume fish, particularly for young or pregnant individuals.

First off, I would like to compliment your choice of a gender-neutral term when referring to pregnant people. Thank you! I am only picking at this sentence because the first two times I read through it, it felt moderately unclear whether you are referring to young or pregnant fish or young or pregnant humans. I broke the sentence into two parts as a proposed edit, though if you are very attached to keeping it as one long sentence, I would consider switching the words ‘individual’ and ‘human’ in their placement in the sentence. I know what you mean, but someone unfamiliar with this topic could get the impression that consuming young or pregnant fish carries the greatest health risk.

Line 93 “organic methylmercury (MeHg) (Bloom, 1992)”

Line 94 after “most bioactive” may consider citing a source here or if also from Bloom, 1992, just cite this at the end of the sentence

Line 108 “none of these occurred in our study area” leads the reader to wonder “these what? (characteristics observed or something else)?” upon rereading I realized the intended meaning may have been “none of these studies was conducted in our study area” (which is what your study is uniquely contributing, so you’ll want to make that clear to emphasize its value)

Line 108 Proposed change: Previous studies, …although none of these studies were conducted in our region. End sentence. New sentence: “Only one of the aforementioned studies examined Hg in lionfish across the entire range of sizes currently being harvested” (cite this one alone so readers can find it and compare).

Line 110 “spatial coverage to include an (not and) assessment of Hg in lionfish”

Line 114 “Impacts on (lionfish) health and reproduction.” For the lionfish or the people who eat them could be made clear.

Line 144 “and not available for some tissue samples collected in the field”

What do you mean by this? As in, some fish were not retrieved and brought back to the lab but instead tissue was sampled from them without removal of the whole fish? If you keep this sentence, you may want to elaborate. The other text of your method section implies that all fish were captured whole during the derby. If this is not the case, I would consider adding detail about what other types of tissue may have been collected and how.

Line 146 Good detail! Left side of fish fillet used to represent edible portion. Excellent to include as would help others who wanted to conduct a similar study for their region do so with same methodology. Also good because readers know the part of fish measured is the part people consume.

Line 149 How did you know the samples were completely dry after 48 hours? Constant weight? You have previously established that this drying procedure is effective? May want to tell the reader why you have confidence in this methodology.

Line 154 comma after; Joiris, 2000;

Line 185 I am curious why you chose to use CRM, NIST Standard Ref 2709a San Joaquin soil as a standard reference material. The other QC/QA procedures all seemed totally reasonable for the DMA-80 calibration and it sounds like you had good reproduceability. My only concern would be if you had poor recovery from fish tissue (while perfectly fine recovery from soil) you would not know this from the soil standard or from agreeable replicates of fish tissue (in the event they both had similarly poor recovery).

I do not think this renders the study unpublishable, just wondering why you did not choose a tissue CRM. If your lab previously demonstrated excellent agreement between a fish tissue standard and the soil standard in use, for example, I would consider providing a figure (like a 1:1 plot) to illustrate that fact as part of supplemental data to dispel any reader skepticism regarding the validity of the measurements.

Line 229 “was independent of fish size, sex (add comma) and location.”

Line 261 concentrations is misspelled (a and r are switched)

Line 291-294 “Mean THg in lionfish (0.05 – 0.06 μg g-1 ww; Table 1) in our study were within the range previously reported from Florida (0.02 – 0.15 μg g-1 ww; Huge et al., 2013; Tremain and O’Donnell, 2014).

I would recommend ending that sentence and starting a new one with something to the effect of: Our study also found mean THg in lionfish to be similar to (or greater than) those studied in Jamaica (0.037 μg g-1 ww; Hoo Fung et al., 2013) and Curaçao (Range: 0.008 – 0.016 μg g-1 ww; Ritger et al. 2018).

I suggest this because 1) 0.037 is not a range and 2) 0.05 – 0.06 is not in the range of 0.008-0.016. To set up the sentence as it currently stands just makes me wonder if the other numbers are typos or if the range is missing for Jamaica (as your stated values are also greater than this).

Line 299 change “more the” to “more than 8-10 times higher”

Line 359-360 “this finding is (in) agreement” ‘in’ is missing

Line 366 “can vary as a function (of) fish size” ‘of’ is missing

Line 371 distribution, detoxification, (and) sequestration in this species

Line 381 “artifact of the data” If you keep this statement, I would elaborate on what you mean specifically.

The way it reads currently led me to have the following questions:

Do male fish have less adipose tissue than female fish and did this pose sampling challenges in regards to uniformity?

Is male fish body fat distribution in a different pattern than for females or is it possible that males burn off and replace body fat throughout their lifetimes, therefore disconnecting the THg stored in adipose tissue from the fish’s other tissues? Perhaps female fish adipose tracks well with liver and muscle because all three increase with fish age whereas male fish have a disconnect here?

I understand that all of these questions are beyond the scope of your current study (which you allude to in the next sentence that calls for more research). However, it might be good to give a specific example of a feature that could have led to a data artifact, such as sample size or a particular limitation of the tissue sampling method.

Line 432 and 435 you are using a comma between 2 authors

Line 435 extra space after Sonne C.

Line 450 you use ‘and’ between 2 authors

I would pick one format and use it throughout.

Line 453 and 456 have comma between 2 authors

Line 468 and Line 485 no comma between single author and date

Line 488 you use a comma after Buddo, DSA. Pick one style for this also.

Line 501 comma missing after Sturup S

Line 506 Period missing after USA

Line 512 subtropical does not need to be capitalized if tropical is not

Line 517 “Enviorn” need to become “Environ”

Line 525 Is Mexico intentionally italicized?

Line 529-530 “2016” is listed prior to the journal title (unlike other entries)

Line 526 add “-“ between page numbers

Line 533 period is missing after page number range

Line 549 “966” not justified like others

Line 553 period missing after 345

Line 564 here the date needs to move to line 566

Line 576 here you list all authors whereas other references are listed as et al. after the first 6 names…I would make all references complete like this one

Line 579 list page range, same format as others

Lines 595-604 text is grey rather than black

Line 596 period missing after 22-32

Line 595 and 598 the year is not formatted like other entries

Line 604 is (8) a volume or issue number?

Line 628 and 632 the year is not formatted like others

Line 649 period missing after page range

Line 668 missing dash in page range

Line 744 the word Ranges does not need to be capitalized

Line 747 put the '1' in parentheses like other citations

Line 755 and 785 you used comma after author last name before initials

Line 778 no comma used between last name and initials …pick one style

Line 790 period missing after page numbers

Reviewer #3: I found the study itself to be interesting and novel. It assesses whether lionfish can be safely consumed by people as a food source, but also as a way to cull the population, since they are invasive and ecologically harmful. I found the study design to be appropriate and scientifically sound and the methods (both analytical and statistical) were appropriate and rigorous.

6. PLOS authors have the option to publish the peer review history of their article (what does this mean?). If published, this will include your full peer review and any attached files.

Reviewer #1: No

Reviewer #2: **Yes: **Jeannette Isabella Calvin

Reviewer #3: No

---

## [Author Response · Author response to Decision Letter 0]

8 Jan 2021

Additional Editor Comments:

In your revision please make sure to fully address each of the reviewers' comments.

In particular for reviewer 1, make sure to address this general criticism of the paper:

"But unfortunately measuring only total Hg does not allow for new insights into the disposition of mercury. And no attention is paid to possible sources of Hg that may explain the observed regional differences. In the introduction broad sweeping objectives are presented for this project, but almost none of those are addressed in the results and discussion." and for Reviewer 2 please indicate why a fish tissue SRM was not used.

 Journal Requirements:

Response: The formatting on the revised manuscript were revised to meet PLOS ONE’s style requirements. 

2. In your Methods section, please provide additional location information of the study sites, including geographic coordinates for the data set if available.

Response: The study sites shown in Fig 1 are approximate, but representative, locations of the general areas in which lionfish were captured. The study utilized recreational divers participating in day long lionfish tournaments to collect lionfish and specific locations of capture (i.e., lat, long) was not provided. 

Response: The relevant accession numbers or DOIs necessary to access the data will be provided following acceptance.

4. We note that Figure 1 in your submission contain map images which may be copyrighted. All PLOS content is published under the Creative Commons Attribution License (CC BY 4.0), which means that the manuscript, images, and Supporting Information files will be freely available online, and any third party is permitted to access, download, copy, distribute, and use these materials in any way, even commercially, with proper attribution. For these reasons, we cannot publish previously copyrighted maps or satellite images created using proprietary data, such as Google software (Google Maps, Street View, and Earth). For more information, see our copyright guidelines: http://journals.plos.org/plosone/s/licenses-and-copyright.

4.1. You may seek permission from the original copyright holder of Figure 1 to publish the content specifically under the CC BY 4.0 license.

4.2. If you are unable to obtain permission from the original copyright holder to publish these figures under the CC BY 4.0 license or if the copyright holder’s requirements are incompatible with the CC BY 4.0 license, please either i) remove the figure or ii) supply a replacement figure that complies with the CC BY 4.0 license. Please check copyright information on all replacement figures and update the figure caption with source information. If applicable, please specify in the figure caption text when a figure is similar but not identical to the original image and is therefore for illustrative purposes only.

Response: The original figure which was a modified version of a previously published figure (PeerJ) was deleted. The new figure (Fig 1) was modified from a base map created using the USGS National Map Viewer (public domain): http://viewer.nationalmap.gov/viewer/ as suggested above.

Reviewers' comments:

Reviewer's Responses to Questions

Comments to the Author

1. Is the manuscript technically sound, and do the data support the conclusions?

Reviewer #1: Partly

Reviewer #2: Yes

Reviewer #3: Yes

2. Has the statistical analysis been performed appropriately and rigorously?

Reviewer #1: Yes

Reviewer #2: Yes

Reviewer #3: Yes

3. Have the authors made all data underlying the findings in their manuscript fully available?

Reviewer #1: Yes

Reviewer #2: No

Reviewer #3: Yes

4. Is the manuscript presented in an intelligible fashion and written in standard English?

Reviewer #1: Yes

Reviewer #2: Yes

Reviewer #3: Yes

5. Review Comments to the Author

Reviewer #1: This manuscript describes the concentrations of total mercury in lionfish from 2 locations off the Atlantic coast of Florida.

The study appears to be well executed, but is very narrow in its conclusions, and does not add any new insights to the scientific knowledge about mercury accumulation in fish species. The results are relevant for the developing market for this species, confirming what others have published already. But unfortunately measuring only total Hg does not allow for new insights into the disposition of mercury. And no attention is paid to possible sources of Hg that may explain the observed regional differences. In the introduction broad sweeping objectives are presented for this project, but almost none of those are addressed in the results and discussion.

Response: Here we provide a summary of revisions to address the reviewers broad overarching comments above (with accompanying highlighted text from the revised manuscript); however detailed responses also follow individual comments reviewer comments below. We have removed the broad sweeping objectives from the revised manuscript and instead focused primarily on THg concentrations in muscle and their potential impact on human health in a developing fishery market for this species. 

“The goal of this study was to quantify total mercury (THg) in lionfish as a function of capture location, sex, size and tissue type; information that is critical for evaluating the potential risk to consumers of this species in a rapidly developing fishery.” 

THg in this context (human health) is informative as numerous studies demonstrate THg in fish muscle is predominantly methylmercury. Our study expands our knowledge of mercury accumulation in this species by sampling in previously unsampled areas and by increasing the range of sizes (particularly large fish which are most likely to have high mercury) for which information is available. 

Previous studies have provided assessments for mercury risk in lionfish (Hoo Fung et al., 2013; Huge et al., 2014; Tremain and O’Donnell, 2014; Ritger et al., 2018); however, none of these studies were conducted in our region of study. Moreover, only one of the aforementioned studies examined mercury in lionfish across the entire range of sizes currently being harvested (Hoo Fung et al., 2013). 

The utility of the results of THg in other tissues is more limited, as the reviewer states. For these results, we provide a clear discussion and acknowledgment of the study limitations imposed by measuring only THg. 

A key limitation of the current study was that methylmercury was not quantified. While numerous studies indicate that THg is a good proxy for methylmercury in fish muscle (Grieb, 1990; Bloom, 1992; Mieiro et al., 2009; Harley et al. 2015), the fraction of the THg pool present as methylmercury is highly variable in other tissue types (Houserova et al., 2006), can vary as a function of fish size (Joiris et al., 2000) and among species (Mieiro et al., 2009; Berzas Nevado et al., 2011). Future work on Hg speciation in lionfish organ tissues is needed to assess the value of using lionfish as an indicator species to assess ecosystem health and risk (e.g., Havelková et al., 2008), and for better understanding ecophysiological mechanisms underlying Hg distribution, detoxification, and sequestration in this species (Cizdziel et al., 2003). Despite these limitations, THg represents the maximum possible concentration for methylmercury and observed THg in all tissues were below accepted thresholds for negative health effects (Depew et al., 2012) and reproductive impairment (Crump and Trudeau, 2008). Thus, lionfish appear unlikely to be substantially affected by mercury toxicity.

A more detailed discussion of the potential drivers of regional differences is also included the revised manuscript. 

Some detailed comments:

Line 52: a more accurate statement is that in the SE USA there is a species continuum for these two species, or even just one hybrid species (see Wilcox CL, Motomura H, Matsunuma M, Bowen BW. Phylogeography of Lionfishes (Pterois) Indicate Taxonomic Over Splitting and Hybrid Origin of the Invasive Pterois volitans. J Hered. 2018 Feb 14;109(2):162-175). Based on my own observations it is difficult to separate the two species on morphological characteristics, and in one location individuals with characteristics of both species can be found.

Response: We have incorporated this suggested change and added the suggested reference.

A marine invader of particular concern in the western Atlantic Ocean is the Indo-Pacific lionfish (a species continuum of two morphologically indistinct species, Pterois miles and P. volitans or potentially a hybrid of the two; Wilcox et al., 2018). 

Line 59: recently a disease outbreak has shown that the species is not all that resistant (Precipitous Declines in Northern Gulf of Mexico Invasive Lionfish Populations Following the Emergence of an Ulcerative Skin Disease. Harris HE, Fogg AQ, Allen MS, Ahrens RNM, Patterson WF 3rd.Sci Rep. 2020 Feb 4;10(1):1934. doi: 10.1038/s41598-020-58886-8)

Response: We have incorporated this suggested change and added the suggested reference.

Lionfish have long venomous spines that deter predation by native predators (Mumby et al., 2011) and are resistant to common parasites (Sikkel et al., 2014); however a recently discovered outbreak of an ulcerative skin disease in lionfish in the Gulf of Mexico suggests some susceptibility to disease (Harris et al., 2020). 

Line 65/66: However, others have found no significant effects of lionfish on native fish populations (Invasive lionfish had no measurable effect on prey fish community structure across the Belizean Barrier Reef. Hackerott S, Valdivia A, Cox CE, Silbiger NJ, Bruno JF. PeerJ. 2017 May 25;5:e3270. doi: 10.7717/peerj.3270. eCollection 2017.)

Response: We have incorporated this suggested change and added the suggested reference.

Lionfish have been shown to reduce native fish recruitment (Albins and Hixon, 2008) and overall native species biomass (Green et al., 2012) in some studies, but not in others (Hackerott et al., 2017). 

Line 72: however, it is doubtful that spear fish derbies will have a significant effect on populations size (Effectiveness of removals of the invasive lionfish: how many dives are needed to deplete a reef? Usseglio P, Selwyn JD, Downey-Wall AM, Hogan JD. PeerJ. 2017 Feb 23;5:e3043. doi: 0.7717/peerj.3043. eCollection 2017)

Response: We have incorporated this suggested change and added the suggested reference.

Such efforts are being actively promoted by management agencies throughout the western Atlantic Ocean and Caribbean Sea, but require careful implementation to be most effective (Usseglio et al. 2017). 

Line 87: I would say “higher trophic levels” because lionfish are not top predators like sharks, and there food is in general smaller fish.

Response: We have incorporated this suggested change.

Line 101 and further: these numbered aims are laudable, but they are hardly at all discussed in the discussion. For instance: there are several papers on lionfish as biomonitoring species for other chemicals, but none of them is discussed. Aims 2 and 3 are also not discussed, only 4.

Response: We have incorporated this suggested change to focus on the primary goal of the study.

The goal of this study was to quantify total mercury (THg) in lionfish as a function of capture location, sex, size and tissue type; information that is critical for evaluating the potential risk to consumers of this species in a rapidly developing fishery. 

Line 111: Do you really think that Hg in fish is linked to “high coastal human population density”? For as far as I know, Hg sources along the East Coast are mostly atmospheric deposition of elemental Hg, getting in the atmosphere through coal-fired power plants. In coastal anoxic marsh sediments this elemental Hg is turned into organic Hg, which is bioavailable. In the discussion it is suggested that the Everglades are a source of Hg in fish in the Keys, which makes sense to me. But why would atmospheric deposition in North Florida be different from the South? Where are the coal fired power plants located in FL?

Response: We have changed the wording of the sentence to remove confusion to remove the phrase “high coastal human population density”. The rationale for including “high coastal human population density” as a qualifier was not that these areas are likely to have high levels of THg, but that these areas have many people and high levels of recreational fishing effort making the lack of information on THg in these areas important from a public health perspective. The regional differences we observed (SEF vs NEF) and possible explanations are detailed in the discussion (and below). 

Line 148: I don’t think you need to state that only females have ovaries

Response: We have incorporated this suggested change.

For a subset of male (n=26) and female fish (n=31), we collected additional samples of liver, adipose and ovarian tissue for analysis. 

Line 236/237: if you set your significance level a 5%, and there are no significant differences between the sexes, then don’t say that the females are generally higher because it is not significant!

Response: We believe that the two statements are not incongruous. THg concentrations may be not significantly different between sexes and still generally higher in females than in males for a given size. The difference becomes larger with size, which we believe results from sex-specific differences in growth rates (growth dilution) which are well documented for this species. The statement is not crucial to the manuscript, so I will remove it if the editor concurs with the reviewer.

Line 284: point 4) see above: if your statistical analysis says there are no differences, then there are no differences!

Response: See above. Assuming the most likely difference between groups is 0 following a statistical test is a common problem in interpreting statistical output. The most likely estimate of the difference between the groups is the observed difference (a confidence interval for which will include 0). Again, the statement is not crucial to the manuscript, so I will remove it if the editor concurs with the reviewer.

Line 316-323: this is called growth dilution

Response: We have incorporated this suggested change.

Low THg in lionfish may be at least partially explained by growth dilution, given the extremely rapid growth rates of lionfish relative to native predators (Zapp-Sluis et al., 2013). 

Line 337/338: it would be nice to just mention here what these “known processes” are!

Response: We have incorporated more discussion into possible processes causing regional differences in THg.

Higher concentrations of THg in SEF could be linked to ambient oceanographic conditions such as warmer ocean temperatures, which have been shown to increase methylation rates by marine primary producers (Lee and Fisher, 2016). Previous work has noted much larger differences among regions in Florida (Tremain and O’Donnell, 2014); however that study examined mercury at a larger spatial scale, in both nearshore and offshore sites, and sampled in locations with both historic point sources (medical waste incineration) and biogeochemical factors that result in enhanced mercury methylation and bioavailability (Chen et al., 2009; Sparling, 2009; Driscoll et al., 2012) distinct from our offshore collection sites. 

Line 363: I fully agree with this statement!

Reviewer #2: *3. Have the authors made all data underlying the findings in their manuscript fully available?

I only said no because on the PDF I received the statement as to where the data can be found is incomplete: "All data files are available from the University of North Florida database (accession number(s) XXX, XXX.)" X's need to be replace with numbers for reader to be able to access. (Maybe they will be on publication, if so, disregard this comment.)

Response: The relevant accession numbers or DOIs necessary to access the data will be provided following acceptance as required by the journal.

4. Is the manuscript presented in an intelligible fashion and written in standard English?

I answered 'yes' because the edits are minimal. Please see the following comments broken down line by line for typos to fix and minor edits for sentence clarity.

Comments to author:

I realized a bit too late that I wasn’t supposed to provide copyediting along with other review. I am still going to share some details I noticed because I hope they will be useful in assembling the final manuscript. I would like to see this paper published so that others can access these results for comparison and for public knowledge. I appreciate the useful impacts of this study for invasive species control, human health knowledge, and for promoting the idea that lionfish could be consumed instead of other imperiled species.

Line 54 comma after Hamner et al.

Line 63 add comma after Johnston et al.

Line 64 add comma after Cote et al.

Line 65 add comma after Peake et al.

Line 68 add comma after Chagaris et al.

Line 75 add comma after Dahl et al.

Response: We have incorporated these suggested edits and appreciate the level of detail with which reviewer 2 edited the manuscript.

Line 88-92 The accumulation of mercury can directly impact fish health (Adams et al., 2010) and can adversely affect many aspects of reproduction (Weis, 2009). Mercury also poses a serious exposure risk for humans who consume fish, particularly for young or pregnant individuals.

First off, I would like to compliment your choice of a gender-neutral term when referring to pregnant people. Thank you! I am only picking at this sentence because the first two times I read through it, it felt moderately unclear whether you are referring to young or pregnant fish or young or pregnant humans. I broke the sentence into two parts as a proposed edit, though if you are very attached to keeping it as one long sentence, I would consider switching the words ‘individual’ and ‘human’ in their placement in the sentence. I know what you mean, but someone unfamiliar with this topic could get the impression that consuming young or pregnant fish carries the greatest health risk.

Response: We have incorporated this suggested edit.

The accumulation of mercury can not only directly impact fish health (Adams et al., 2010) and adversely affect many aspects of reproduction (Weis, 2009). Mercury also poses a serious exposure risk for humans who consume fish, particularly for young or pregnant individuals (NRC, 2000; U.S. EPA, 2001; Karagas et al., 2012). 

Line 93 “organic methylmercury (MeHg) (Bloom, 1992)”

Line 94 after “most bioactive” may consider citing a source here or if also from Bloom, 1992, just cite this at the end of the sentence

Response: We have incorporated this suggested edit.

Further, most of the mercury present in fish muscle is present as organic methylmercury (MeHg) (Bloom, 1992; Mieiro et al., 2009; Harley et al. 2015), the most highly toxic and bioactive form (Mergler et al., 2007). 

Line 108 “none of these occurred in our study area” leads the reader to wonder “these what? (characteristics observed or something else)?” upon rereading I realized the intended meaning may have been “none of these studies was conducted in our study area” (which is what your study is uniquely contributing, so you’ll want to make that clear to emphasize its value)

Response: We changed the text for clarification.

Previous studies have provided assessments for mercury risk in lionfish (Hoo Fung et al., 2013; Huge et al., 2014; Tremain and O’Donnell, 2014; Ritger et al., 2018); however, none of these studies were conducted in our region of study. 

Line 108 Proposed change: Previous studies, …although none of these studies were conducted in our region. End sentence. New sentence: “Only one of the aforementioned studies examined Hg in lionfish across the entire range of sizes currently being harvested” (cite this one alone so readers can find it and compare).

Response: We have incorporated this suggested edit.

Previous studies have provided assessments for mercury risk in lionfish (Hoo Fung et al., 2013; Huge et al., 2014; Tremain and O’Donnell, 2014; Ritger et al., 2018); however, none of these studies were conducted in our region of study. Moreover, only one of the aforementioned studies examined mercury in lionfish across the entire range of sizes currently being harvested (Hoo Fung et al., 2013). 

Line 110 “spatial coverage to include an (not and) assessment of Hg in lionfish”

Response: We have incorporated this suggested edit.

Line 114 “Impacts on (lionfish) health and reproduction.” For the lionfish or the people who eat them could be made clear.

Response: We changed the text for clarification.

This study builds on earlier work by (1) expanding the spatial coverage to include an assessment of mercury in lionfish from unstudied regions, (2) expanding the range of sizes examined (particularly large individuals which have the highest potential for mercury bioaccumulation), and (3) quantifying mercury levels in different lionfish tissues to assess potential impacts on lionfish health and reproduction

Line 144 “and not available for some tissue samples collected in the field”

What do you mean by this? As in, some fish were not retrieved and brought back to the lab but instead tissue was sampled from them without removal of the whole fish? If you keep this sentence, you may want to elaborate. The other text of your method section implies that all fish were captured whole during the derby. If this is not the case, I would consider adding detail about what other types of tissue may have been collected and how.

Response: We changed the text to add more detail. 

Sex determination was not possible for many smaller immature individuals (Morris et al., 2012) and not available for a limited number of lionfish (n=5) which sustained extensive damage during capture and field processing.

Line 146 Good detail! Left side of fish fillet used to represent edible portion. Excellent to include as would help others who wanted to conduct a similar study for their region do so with same methodology. Also good because readers know the part of fish measured is the part people consume.

Response: As suggested, no change was made.

Line 149 How did you know the samples were completely dry after 48 hours? Constant weight? You have previously established that this drying procedure is effective? May want to tell the reader why you have confidence in this methodology.

Response: We changed the text to add more detail. 

All samples were dried in a 60°C oven for 48 hours (a duration sufficient to achieve constant weight of samples) then homogenized using a mortar and pestle prior to analysis of total mercury concentrations (hereafter THg). 

Line 154 comma after; Joiris, 2000;

Response: We have incorporated this suggested edit.

Line 185 I am curious why you chose to use CRM, NIST Standard Ref 2709a San Joaquin soil as a standard reference material. The other QC/QA procedures all seemed totally reasonable for the DMA-80 calibration and it sounds like you had good reproduceability. My only concern would be if you had poor recovery from fish tissue (while perfectly fine recovery from soil) you would not know this from the soil standard or from agreeable replicates of fish tissue (in the event they both had similarly poor recovery).

I do not think this renders the study unpublishable, just wondering why you did not choose a tissue CRM. If your lab previously demonstrated excellent agreement between a fish tissue standard and the soil standard in use, for example, I would consider providing a figure (like a 1:1 plot) to illustrate that fact as part of supplemental data to dispel any reader skepticism regarding the validity of the measurements.

Response: The reviewer raises a valid point. Unfortunately, at the time of the study we attempted to obtain fish tissue CRMs, however they were not available for purchase from vendors. Thus, we purchased the Soil CRM. We have recently acquired fish tissue CRM, but have not yet conducted the suggested analysis. 

Line 229 “was independent of fish size, sex (add comma) and location.”

Response: We have incorporated this suggested edit.

Line 261 concentrations is misspelled (a and r are switched)

Response: We have incorporated this suggested edit.

Line 291-294 “Mean THg in lionfish (0.05 – 0.06 μg g-1 ww; Table 1) in our study were within the range previously reported from Florida (0.02 – 0.15 μg g-1 ww; Huge et al., 2013; Tremain and O’Donnell, 2014).

I would recommend ending that sentence and starting a new one with something to the effect of: Our study also found mean THg in lionfish to be similar to (or greater than) those studied in Jamaica (0.037 μg g-1 ww; Hoo Fung et al., 2013) and Curaçao (Range: 0.008 – 0.016 μg g-1 ww; Ritger et al. 2018).

I suggest this because 1) 0.037 is not a range and 2) 0.05 – 0.06 is not in the range of 0.008-0.016. To set up the sentence as it currently stands just makes me wonder if the other numbers are typos or if the range is missing for Jamaica (as your stated values are also greater than this).

Response: We have incorporated this suggested edit to add clarification. There was also a typo: 0.016 was 0.106 for THg in Curacao, which is the range of our results.

Mean THg in lionfish (0.05 – 0.06 μg g-1 ww; Table 1) in our study were within the range previously reported from Florida (0.02 – 0.15 μg g-1 ww; Huge et al., 2013; Tremain and O’Donnell, 2014). Our results were also within the range of lionfish THg studied in Jamaica (0.016 – 0.061 μg g-1 ww; Hoo Fung et al., 2013) and Curaçao (Range: 0.008 – 0.106 μg g-1 ww; Ritger et al. 2018). 

Line 299 change “more the” to “more than 8-10 times higher”

Response: We have incorporated this suggested edit.

Line 359-360 “this finding is (in) agreement” ‘in’ is missing

Response: We have incorporated this suggested edit.

Line 366 “can vary as a function (of) fish size” ‘of’ is missing

Response: We have incorporated this suggested edit.

Line 371 distribution, detoxification, (and) sequestration in this species

Response: We have incorporated this suggested edit.

Line 381 “artifact of the data” If you keep this statement, I would elaborate on what you mean specifically.

The way it reads currently led me to have the following questions:

Do male fish have less adipose tissue than female fish and did this pose sampling challenges in regards to uniformity?

Is male fish body fat distribution in a different pattern than for females or is it possible that males burn off and replace body fat throughout their lifetimes, therefore disconnecting the THg stored in adipose tissue from the fish’s other tissues? Perhaps female fish adipose tracks well with liver and muscle because all three increase with fish age whereas male fish have a disconnect here?

I understand that all of these questions are beyond the scope of your current study (which you allude to in the next sentence that calls for more research). However, it might be good to give a specific example of a feature that could have led to a data artifact, such as sample size or a particular limitation of the tissue sampling method.

Response: We have removed this statement as suggested by the reviewer.

Line 432 and 435 you are using a comma between 2 authors

Line 435 extra space after Sonne C.

Line 450 you use ‘and’ between 2 authors

I would pick one format and use it throughout.

Line 453 and 456 have comma between 2 authors

Line 468 and Line 485 no comma between single author and date

Line 488 you use a comma after Buddo, DSA. Pick one style for this also.

Line 501 comma missing after Sturup S

Line 506 Period missing after USA

Line 512 subtropical does not need to be capitalized if tropical is not

Line 517 “Enviorn” need to become “Environ”

Line 525 Is Mexico intentionally italicized?

Line 529-530 “2016” is listed prior to the journal title (unlike other entries)

Line 526 add “-“ between page numbers

Line 533 period is missing after page number range

Line 549 “966” not justified like others

Line 553 period missing after 345

Line 564 here the date needs to move to line 566

Line 576 here you list all authors whereas other references are listed as et al. after the first 6 names…I would make all references complete like this one

Line 579 list page range, same format as others

Lines 595-604 text is grey rather than black

Line 596 period missing after 22-32

Line 595 and 598 the year is not formatted like other entries

Line 604 is (8) a volume or issue number?

Line 628 and 632 the year is not formatted like others

Line 649 period missing after page range

Line 668 missing dash in page range

Line 744 the word Ranges does not need to be capitalized

Line 747 put the '1' in parentheses like other citations

Line 755 and 785 you used comma after author last name before initials

Line 778 no comma used between last name and initials …pick one style

Line 790 period missing after page numbers

Response: We have incorporated each of these suggested edits, and very much appreciate the detail with which reviewer 2 edited the document. These edits were quite helpful in generating the revised manuscript.

Reviewer #3: I found the study itself to be interesting and novel. It assesses whether lionfish can be safely consumed by people as a food source, but also as a way to cull the population, since they are invasive and ecologically harmful. I found the study design to be appropriate and scientifically sound and the methods (both analytical and statistical) were appropriate and rigorous.

---

## [Editor Report · Decision Letter 1]

13 Jan 2021

PONE-D-20-15978R1

Total mercury concentrations in invasive lionfish (Pterois volitans/miles) from the Atlantic coast of Florida

PLOS ONE

Dear Dr. Johnson,

Thank you for submitting your manuscript to PLOS ONE. After careful consideration, we feel that it has merit but does not fully meet PLOS ONE’s publication criteria as it currently stands. Therefore, we invite you to submit a revised version of the manuscript that addresses the points raised during the review process.

We look forward to receiving your revised manuscript.

Kind regards,

Myra E Finkelstein

Academic Editor

PLOS ONE

Additional Editor Comments (if provided):

One major concern with the QAQC of the mercury data was that the authors used an inappropriate CRM to assess the accuracy of total mercury measurements. This was pointed out by one of the reviewers on the prior draft. In response to this comment the authors stated in this revision: "The reviewer raises a valid point. Unfortunately, at the time of the study we attempted to obtain fish tissue CRMs, however they were not available for purchase from vendors. Thus, we purchased the Soil CRM. We have recently acquired fish tissue CRM, but have not yet conducted the suggested analysis."

As the main findings and conclusions of this paper are based on total mercury measurements, before the revised paper is sent back out to review, I think it is necessary to run the fish tissue CRM that has been purchased (which should be a quick and simple thing to do to) compare with the soil CRM as the reviewer suggested.

Comment from the reviewer on the prior draft: "If your lab previously demonstrated excellent agreement between a fish tissue standard and the soil standard in use, for example, I would consider providing a figure (like a 1:1 plot) to illustrate that fact as part of supplemental data to dispel any reader skepticism regarding the validity of the measurements.”

---

## [Author Response · Author response to Decision Letter 1]

8 Jun 2021

This is the second revision of the manuscript. In this second revision, we specifically address reviewer/editor concerns relative to our use of a solid soil Certified Reference Material (CRM) and not a fish tissue CRM. As suggested, we have conducted supplemental paired analysis of Hg using both CRMs. These analyses demonstrated that both our original standard CRM and the fish protein CRM (DORM-4) were within expected ranges. Specifically, average DORM-4 concentrations for Hg were 0.38 ug/g (expected reference range: 0.41 +/- 0.055 ug/g).

---

## [Decision Letter · Decision Letter 2]

13 Jul 2021

Total mercury concentrations in invasive lionfish (Pterois volitans/miles) from the Atlantic coast of Florida

PONE-D-20-15978R2

Dear Dr. Johnson,

We’re pleased to inform you that your manuscript has been judged scientifically suitable for publication and will be formally accepted for publication once it meets all outstanding technical requirements.

Kind regards,

Myra E Finkelstein

Academic Editor

PLOS ONE

Additional Editor Comments (optional):

Reviewers' comments:

Reviewer's Responses to Questions

**Comments to the Author**

1. If the authors have adequately addressed your comments raised in a previous round of review and you feel that this manuscript is now acceptable for publication, you may indicate that here to bypass the “Comments to the Author” section, enter your conflict of interest statement in the “Confidential to Editor” section, and submit your "Accept" recommendation.

Reviewer #1: All comments have been addressed

Reviewer #2: All comments have been addressed

2. Is the manuscript technically sound, and do the data support the conclusions?

Reviewer #1: Yes

Reviewer #2: Yes

3. Has the statistical analysis been performed appropriately and rigorously? 

Reviewer #1: Yes

Reviewer #2: Yes

4. Have the authors made all data underlying the findings in their manuscript fully available?

Reviewer #1: Yes

Reviewer #2: Yes

5. Is the manuscript presented in an intelligible fashion and written in standard English?

Reviewer #1: Yes

Reviewer #2: Yes

6. Review Comments to the Author

Reviewer #1: (No Response)

Reviewer #2: Excellent! Thank you for including the fish tissue standard information. The sections rewritten for clarity improved the readability of the manuscript. It is now even more informative and valuable due to the inclusion of specific details (i.e. description of historic point sources/historic incineration of medical waste, reason as to why the sex could not be determined for certain individuals).

7. PLOS authors have the option to publish the peer review history of their article (what does this mean?). If published, this will include your full peer review and any attached files.

Reviewer #1: No

Reviewer #2: **Yes: **Nettie Calvin

---

## [Editor Report · Acceptance letter]

9 Sep 2021

PONE-D-20-15978R2 

Total mercury concentrations in invasive lionfish (*Pterois volitans/miles*) from the Atlantic coast of Florida 

Dear Dr. Johnson:

I'm pleased to inform you that your manuscript has been deemed suitable for publication in PLOS ONE. Congratulations! Your manuscript is now with our production department. 

Kind regards, 

on behalf of

Dr. Myra E Finkelstein 

Academic Editor

PLOS ONE